

# Identification of secondary aerosol precursors emitted by an aircraft turbofan

Dogushan Kilic[1], Imad El Haddad[1], Benjamin T. Brem[2,5], Emily Bruns[1], Carlo Bozetti[1], Joel Corbin[1], Lukas Durdina[2,5], Ru-Jin Huang[1], Jianhui Jiang[1], Felix Klein[1], Avi Lavi[4], Simone M. Pieber[1], Theo Rindlisbacher[3], Yinon Rudich[4], Jay G. Slowik[1], Jing Wang[2,5], Urs Baltensperger[1], and Andre S. H. Prévôt[1]

[1]Laboratory of Atmospheric Chemistry, Paul Scherrer Institute, Villigen PSI, 5400, Switzerland

[2]Laboratory for Advanced Analytical Technologies, Empa, Dübendorf, 8600, Switzerland

[3]Federal Office of Civil Aviation, Bern, 3003, Switzerland

[4]Department of Earth and Planetary Sciences, Weizmann Institute of Science - Rehovot – Israel

[5]Institute of Environmental Engineering, ETH Zurich, Zurich, 8093, Switzerland

*Correspondence to:* andre.prevot@psi.ch and imad.el-haddad@psi.ch

**Abstract.** Oxidative processing of aircraft turbine-engine exhaust was studied using a potential aerosol mass (PAM) chamber at different engine loads corresponding to typical flight operations. Measurements were conducted at an engine test cell. Organic gases (OGs) and particle emissions pre/post PAM were measured. A suite of instruments, including a proton-transfer-reaction mass spectrometer (PTR-MS) for OGs, a multi-gas analyzer for CO, $CO_2$, $NO_X$, and an aerosol mass spectrometer (AMS) for non-refractory particulate matter (NR-$PM_1$) were used. Total aerosol mass was dominated by secondary aerosol formation, which was approximately two orders of magnitude higher than the primary aerosol. The chemical composition of both gaseous and particle emissions were also monitored at different engine loads and were thrust dependent. At idling load (thrust 2.5-7%), more than 90% of the secondary particle mass was organic and could be explained by the oxidation of gaseous aromatic species/ OGs; *e.g.* benzene, toluene, xylenes, tri-, tetra-, and pentamethyl-benzene and naphthalene. The oxygenated-aromatics, *e.g.* phenol, furans, were also included in this aromatic fraction and their oxidation could alone explain up to 25% of the secondary organic particle mass at idling loads. The organic fraction decreased with thrust level, while the inorganic fraction increased. At an approximated cruise load sulfates comprised 85% of the total secondary particle mass.

## 1    Introduction

Airport activities emit both particulate and gaseous emissions (Unal et al., 2005; Hudda et al., 2014), and are a significant source of local gas- and particle-phase pollutants (Westerdahl et al., 2008). These emissions affect public health (Lin et al., 2008) and local air quality by increasing pollutant concentrations, *e.g.* ultrafine particulate matter (PM) number concentrations, at the surrounding residential areas (Hudda and Fruin, 2016; Hudda et al., 2016).

The dominant source of airport aerosol is aircraft engine exhaust (Kim, 2009), and is classified as either directly emitted primary aerosol (PA) or secondary aerosol (SA). Due to the high combustion efficiency, PA from aircraft engines contains mainly black carbon (BC) whereas SA is formed by the oxidation of emitted precursor gases such as non-methane organic gases (NMOGs) strongly depend on aircraft engine operating conditions (Kinsey et al., 2010) *e.g.* the BC emission index (EI, g/kg fuel) of a gas-turbine engine is usually higher at cruise climb-out and take-off loads (above 60% of the maximum thrust) than at lower loads used at idle, taxi (7%) and approach (30%) (Liati et al., 2014; Brem et al., 2015). In contrast to BC, NMOG emissions, including *e.g.* aromatic hydrocarbons, aliphatic hydrocarbons and carbonyls, are clearly highest at low loads (Spicer et al., 1994; Slemr et al., 2001; Anderson et al., 2006; Herndon et al., 2006; Kilic et al., 2017).

Aging of fossil fuel combustion exhaust leads to SA/PA ratios higher than 1. Single-ring aromatics are traditionally thought to be the most important secondary organic aerosol (SOA) precursors from combustion emissions. While this has been shown to be the case for some emissions, *e.g.* from 2-stroke engines (Platt et al., 2014), in other cases non-traditional precursors were assessed to be responsible for the bulk of the SOA mass formed, *e.g.* for biomass smoke (Bruns et al., 2016) or on-road vehicles (Platt et al., 2013; 2017; Pieber et al., 2017). Similar to these emissions, aging of aircraft emissions studied by Miracolo et al. (2011; 2012) in a smog chamber produced substantial amounts of secondary PM exceeding primary PM emissions several-fold. The authors showed the dominance of secondary organic aerosol (SOA) at low loads, while at high loads sulfate was the main SA produced. While single-ring aromatic compounds determined using gas-chromatography/mass





spectrometry seemed to be important precursors of the SOA formed, a greater part of SOA was believed to
originate from non-traditional precursors, whose nature remains to be identified (Miracolo et al., 2011; 2012).
In this study, we measured the SA production potential of aircraft jet engine exhaust as a function of engine load
and examined the bulk gas-phase organic emissions and their SOA formation potential. SOA was produced by
OH-initiated oxidation of aircraft NMOG emissions in a potential aerosol mass (PAM) flow reactor (Kang et al.,
2007). Primary and secondary PM mass was characterized for different engine loads, using an aerosol mass
spectrometer (AMS). SOA precursors were analyzed in real-time by a proton-transfer-reaction mass
spectrometer (PTR-MS) and SOA closure was examined under different conditions. The impact of these
emissions and their SOA potential in typical urban atmospheres, at the proximity of airports is assessed and
compared to other mobile sources.
**2  Methods**
**2.1  Experimental setup**
Exhaust measurements were conducted to characterize NMOG and non-refractory submicron particulate mass
(NR-PM$_1$) emissions from an in-production CFM56 variant turbofan in the test cell of SR Technics at Zurich
Airport.  The test engine was fueled with standard JET A-1 fuel, and was operated at several engine loads,
selected to represent aircraft activities during a typical landing/take-off (LTO) cycle. Engine loads were set by
specifying the combustion chamber inlet temperature values which correlate with a specific thrust (lbf) at
standard atmospheric conditions. The selected loads included idle-taxi (3-7% of the maximum thrust), approach
(30% of the maximum thrust), and an approximated cruise load (50-65% of the maximum static thrust). After
starting the engine, a warm-up sequence of 25 minutes ran before each test, consisting of five minute-long steps
at thrusts of 5%, 15%, 7%, 65% and 85% in sequence.
A simplified scheme of the experimental setup is shown in Figure 1 and is discussed in detail elsewhere (Kilic et
al., 2017). Details about the sampling system for non-volatile particle emissions can be found in Durdina et al.
(2017) and Brem et al. (2015). The turbine engine exhaust was sampled by a single-point probe with an inner
diameter of 8 mm, located 0.7 m downstream of the engine exit plane. The exhaust drawn by the probe was
directed through a heated (160°C) transfer line to three different lines: (i) the raw gas line, (ii) diluted emissions
line and (iii) diluted aged emissions line. CO, CO$_2$, and NO$_X$ were measured by a multi-gas analyzer (PG250,
Horiba Inc.) installed on the raw line. On the diluted line, primary gas and particle measurements were
performed. Two ejector dilutors (DEKATI DI-1000) were installed in sequence on this transfer line; after the
first dilution, sampling lines were heated to 120°C. The sample was diluted with synthetic air (99.999% purity)
either by a factor of 10 or 100, depending on the NMOG concentration. The NMOGs were quantified and
characterized by a proton-transfer-reaction time-of-flight mass spectrometer (PTR-ToF-MS) together with a
flame ionization hydrocarbon detector (FID) (APHA 370 THC Monitor). The concentration of equivalent black
carbon (eBC) was determined by a 7-wavelength aethalometer (Drinovec et al., 2015) based on optical
absorption.
Aging of the engine exhaust emissions was achieved by using a potential aerosol mass (PAM) chamber with a
continuous flow of 7.6 l/min and a volume of 13.3 liters. Two mercury lamps (emission lines at wavelengths
λ=182 nm – 254 nm, BHK Inc.), mounted inside the PAM, were used to irradiate HONO and O$_2$ required for
hydroxyl radical (OH) formation. Different time-integrated OH exposures (molecules cm$^{-3}$ h) were achieved by
modulating UV lamp intensity *e.g.* 80%, 90%, 100%. HONO to boost OH concentrations, and D9-butanol to
trace OH exposure (Barmet et al., 2012), were injected with flows of 1.8 and 0.4 l/min, respectively. Further, the
PAM was also humidified (∼20% relative humidity) by injecting synthetic air with water vapor (with a flow of
1.6 l/min). All measurements were conducted at 295-298°K. Secondary aerosol formation was measured after
the PAM, while the primary emissions were measured from the bypass line.
Aging in a PAM is not completely analogous to that in a smog chamber, due to higher oxidant concentrations.
However, intercomparison studies suggest that the amount of SOA production and its bulk elemental
composition are comparable for both single precursors (*e.g.* α-pinene) (Lambe et al., 2015) and complex
emissions (*e.g.* wood combustion) (Bruns et al, 2015). In addition, in both the PAM and chambers, the dominant
oxidation pathways are similar to those in ambient air (Peng et al, 2015; 2016).
**2.2  Instrumentation**
**2.2.1  PTR-ToF-MS**
NMOGs having a higher proton affinity than water were quantified by a PTR-ToF-MS (PTR-TOF 8000,
Ionicon Analytik G.m.b.H., Innsbruck, Austria) (Jordan et al., 2009). NMOG molecules were positively charged
in the ionization unit (drift tube) of the instrument via hydronium ions (H$_3$O$^+$), and the generated ions/fragments
were measured by a time-of-flight mass spectrometer. The PTR-ToF-MS utilized a drift voltage (Udrift) of 550
V, a drift chamber temperature (Tdrift) of 60°C and a drift pressure (pdrift) of 2.2 mbar, maintaining a reduced
electric field (*E/N*) of ∼120 Townsends (Td). Data were collected with one second time resolution.



Tofware post-processing software (version 2.4.5, TOFWERK AG, Thun, Switzerland; PTR module as
distributed by Ionicon Analytik GmbH, Innsbruck, Austria), running in the Igor Pro 6.3 environment
(Wavemetrics Inc., Lake Oswego, OR, USA), was used for data analysis. The ion transmission function,
required to convert counts (cps) to volume mixing ratios (ppbv), was quantified using a gas standard containing
a mixture of 12 compounds (100 ppbv each) spanning mass-to-charge ratios ($m/z$) from $m/z$ 33 to 181 (Carbagas
AG., Zurich, Switzerland). Volume mixing ratios (ppbv) were calculated according to De Gouw and Warneke,
2007, using $H_3O^+$/NMOG reaction rate constants ($k$) from Cappellin et al., 2012, when available, and assuming
$2 \times 10^{-9}$ $cm^3$ $s^{-1}$ otherwise.
During the exothermic proton-transfer reaction, some molecular fragments are formed in the drift chamber, with
the extent of fragmentation depending on the chamber conditions and functional groups in the molecules
(Gueneron et al., 2015). In particular, hydrocarbon fragments are obtained from aldehydes and dehydration of
some oxygenated ions. Assignment of these fragment ions to the corresponding parent ions is important for
quantification. The NMOG mixing ratios were corrected by accounting for this fragmentation. The compounds
were measured based on their parent ions, then their fragments were subtracted based on reference
fragmentation patterns. These subtractions combine a detailed fragmentation table for aldehydes using the
current drift chamber conditions from Klein et al. (2016) and fragmentation patterns for aromatic compounds
measured under similar chamber conditions reported ($E/N$ ~120 Td) in other studies (Buhr et al., 2002; Brown et
al., 2010; Gueneron et al., 2015). The fragmentation of detected compounds containing other functional groups
(e.g. hydrocarbons and non-aldehyde oxygenated compounds) cannot be fully excluded but are not expected to
cause significant error since the observed parent molecules were primarily low molecular weight alcohols and
acids (e.g. methanol, formic and acetic acid) that are less susceptible to fragmentation (de Gouw and Warneke,
2007). The NMOGs were then classified (acids, alcohols, aromatics, non-aromatics hydrocarbons and
unclassified hydrocarbon fragments, nitrogen and sulfur containing compounds, other oxygen containing
compounds, unidentified peaks) according to Kilic et al. (2017).

### 2.2.2 AMS

The condensed phase was continuously monitored before and after the PAM using a high resolution time-of-
flight aerosol mass spectrometer (AMS) and a scanning mobility particle sizer (SMPS). The reader is referred to
DeCarlo et al. (2006) for a more detailed description of the AMS operating principles, calibrations protocols,
and analysis procedures. Briefly, a particle beam sampled through an aerodynamic lens is alternately blocked
and unblocked, yielding the bulk particle mass spectra (MS mode) of the non-refractory (NR) species, including
organic aerosols (OA), $NO_3^-$, $SO_4^{2-}$, $NH_4^+$, and $Cl^-$. The NR particles are flash vaporized by impaction on a
heated tungsten surface (heated to ~ 600°C) at ~ $10^{-7}$ Torr. The resulting gases are ionized by electron
ionization (EI, 70 eV) and the mass-to-charge ratios ($m/z$) of the fragments are determined by the ToF mass
spectrometer. The AMS was operated in the V-mode, with a time resolution of 30 sec. The AMS data were
analyzed using the SQUIRREL (version 1.52L) and PIKA (1.11L) analysis software in Igor Pro 6.3
(WaveMetrics). Standard relative ionization efficiencies (RIE) were assumed for the organic aerosol and
chloride (RIE = 1.4, and 1.3, respectively) and experimentally determined for sulfate and ammonium (RIE =
~1.1 and ~4, respectively). The collection efficiency due to the particle bounce was determined to be ~1 under
our conditions for organic rich aerosols by comparing the AMS mass to the SMPS volume (assuming an OA
density of 1.4).

### 2.3 Data analysis

Emissions from the aircraft turbofan were measured at different thrust levels referred to as "test points"
hereinafter. Test point durations were 18 minutes, except for one 60 minute-long run. Test points were
systematically interspersed with five minute-long periods to clean the PAM and the transfer lines by flushing the
setup with synthetic air. The averaging of the primary emissions started from the third minute of a test point
when the engine operation was stable. After sampling primary emissions for five to eight minutes while
bypassing the PAM, secondary formation was measured after the PAM during the last five to eight minutes of
the test point. This allows SOA to reach a steady state in the PAM. During each test, the PA concentration was
measured by bypassing the PAM, while the SA concentration was calculated by subtracting PA from the OA
measured after aging (after the PAM). Both PA and SA concentrations were determined by AMS.
The expected SOA concentration from the sum of all NMOGs detected by the PTR-ToF-MS was also calculated
by multiplying the NMOGs oxidized in the PAM by its corresponding SOA yields, according to Eq. 1:




$$\sum_{i=1}^{n} SOA\ modelled = \sum_{i=1}^{n} \Delta NMOG_i \times Yield_i \qquad \text{Eq. 1,}$$
where $n$ is the number of NMOGs quantified and $\Delta NMOG$ is the difference between the primary NMOG
concentration and the NMOG concentration after aging. The same approach was applied by Bruns et al. (2016)
and yields used can be found in Table 2. SOA yields available in the literature were used when possible.
Otherwise, SOA yields of 0.2 were assumed as a lower limit estimate for aromatic and oxy-aromatics for which
no SOA-yield values were reported (Presto et al., 2010; Tkacik et al., 2012), similar to Bruns et al. (2016). A
yield of 0.15 was assumed for other NMOGs, including non-aromatic hydrocarbons and carbonyls. As NO is
completely consumed in the PAM, we have chosen yields from low $NO_X$ conditions for aromatic hydrocarbons
(Ng et al., 2007; Chan et al., 2009; Hildebrandt et al., 2009; Nakao et al., 2011). The SOA contribution from
organic gases lighter than benzene ($C_6H_6$) was neglected. Predicted NMOG contributions to SOA are provided
in the Results section.
Emission indices ($EI$, g/kg fuel) were calculated using a mass balance on fuel carbon:
$$EI = [X] \times \left[ \frac{MW_{CO_2}}{MW_C \times \Delta CO_2} + \frac{MW_{CO}}{MW_C \times \Delta CO} \right] \times C_f \qquad \text{Eq. 1,}$$
where $X$ denotes the pollutant concentration ($\mu g/m^3$) and $MW$ (g/mole) is the molecular weight of the species
denoted by the subscript. Background-subtracted CO and $CO_2$ concentrations ($\mu g/m^3$) are denoted as $\Delta CO$ and
$\Delta CO_2$, respectively. $C_f$ is the carbon fraction of the JET-A1 fuel used during the campaign and was measured as
0.857 based on ASTM D 5291 method (ASTM, 1996).
**3      Results and discussion**
**3.1      SOA formation as a function of OH exposure**
The evolution of the chemical composition of the primary organic gases and NR-PM$_1$ components with
increasing OH exposure is shown in Figure 2 for engine idling operation (thrust 3%). Measurements were
conducted for primary emissions, as well as for OH exposures of $59 \times 10^6$, $88 \times 10^6$, and $113 \times 10^6$ molecules cm$^{-3}$
h, which correspond to approximately 39, 58, and 75 hours of atmospheric aging under an average tropospheric
OH concentration of $1.5 \times 10^6$ molecules cm$^{-3}$ (Mao et al., 2009). The OH exposure, calculated using d9-
butanol as a tracer, was varied by varying the light intensity.
Figure 2a shows the OG composition under these conditions with compounds classified as a function of their
molecular composition, as described in Kilic et al. (2017). A stepwise increase of the OH exposure reduced the
NMOG mass detected in the chamber by 35%, 40% and 50%. Except for carboxylic acids, the concentrations of
all NMOGs decreased during aging, indicating that their loss rate exceed their production from other NMOGs.
For example, aromatic compounds and carbonyls were oxidized in the PAM by up to 90% and 50%,
respectively, while the acids doubled after 75 hours of daytime-equivalent aging.
Figure 2b shows a time series of secondary NR-PM$_1$ composition, as well as the concentrations of two of the
most abundant aromatic gases, $C_{10}H_{14}$ and $C_{11}H_{16}$, for the same experiment. Here stable oxidation conditions
were alternated with sampling of primary emissions, with OH exposures indicated in the figure. Secondary
aerosol, especially SOA, dominated the total NRPM$_1$. By increasing the OH exposure from $59 \times 10^6$ to $88 \times 10^6$,
the generated SOA increased by approximately 14%. However, increasing the OH exposure further to $113 \times 10^6$
molecules cm$^{-3}$ h yielded only an additional 3% increase in SOA mass. This suggests that at these OH
exposures, the bulk of SOA precursors have reacted and the additional SOA production did not significantly
exceed its loss. Under these conditions, the formed SOA may be considered as a reasonable estimate for the
total SOA potential. The observed production rate of SOA against OH exposure is consistent with precursor
reaction rates of $8 \times 10^{-12}$ molecule$^{-1}$ cm$^3$ s$^{-1}$. This estimate is based on the assumption of a constant SOA mass
yield with aging and instantaneous equilibrium partitioning of the condensable gases, and is therefore lower than
the reaction rates of the main identified precursors (see below). SOA production rates are thus expected to be
faster in the ambient atmosphere.
**3.2      Particle and gaseous emissions as a function of engine load**
Figure 3 shows both average primary and secondary emissions indices for varying engine loads (left) and EIs
from individual test points (right). The NMOG EI decreased from 30 to 0.8 g/kg fuel when the thrust level
increased from 3-5% to 90%. At thrust 3-5%, the emissions of gaseous aromatic-hydrocarbons were highest
(with an EI of ~5g/kg fuel) and decreased with increasing thrust (with an EI of ~0.15 g/kg fuel at thrust 90%).
Similar to aromatic gases, SOA were formed mostly at 3-5% thrust and had a declining trend with thrust. In
contrast, BC, POA and secondary SO$_4$ EIs were highest during the approximated cruise load (thrust 60%). At



these conditions, secondary NR-PM$_1$ was mostly inorganic and SOA mass was comparable to that of primary
carbonaceous emissions (BC + POA). SOA was approximately 100 times higher than POA at idle and only 10
times higher at cruise (Figure 3). This dependence of the aged aerosol composition on the thrust level, obtained
using the PAM reactor, confirm quite readily the previous results obtained in a smog chamber (Miracolo et al.,
216 2011; 2012).

### 3.3 Precursor gases of SOA: Idling

Single-ring aromatics, such as xylenes, methylbenzenes, toluene, and benzene, were previously linked with
SOA formation (*e.g.*(Odum et al., 1997; Ng et al, 2007)). These aromatic gases are important contributors in the
emissions from combustion sources such as two-stroke scooters (Platt et al., 2013), or wood burning (Bruns et
al., 2016). Idling exhaust contained 20% (mass weighted) of aromatic HCs. Figure 4 presents the mass fractions
of aromatic hydrocarbons in primary exhaust for an idling turbine engine. More than half of the aromatic
hydrocarbons emitted were single-ring aromatics. 75 - 95% of these aromatics were oxidized with an OH
exposure of ~90 x 10$^6$ molecules cm$^{-3}$ h in the PAM.
By using previously reported SOA yields (Table 2) for NMOGs, SOA production was predicted from individual
precursors according to Eq. 1. Figure 5 shows a comparison of the predicted SOA with the SOA determined by
AMS measurements (top) and the predicted SOA contribution by the oxidation of NMOGs in the PAM
(bottom), for two idling thrusts, 2-5% (left) and 6-7% (right). The predicted SOA from the NMOGs reacted are
shown at the bottom panel of the figure and compound class-specific SOA fractions are separated for aromatic
HCs, oxygenated-aromatics, other HCs, N-containing OGs and other OGs.
Results in Figure 5 indicate that the most important SOA precursors emitted by turbine engines at idle are
aromatic hydrocarbons such as benzene derivatives but also oxygenated aromatics such as phenol. The predicted
SOA formed by aromatics alone, both by aromatic hydrocarbons (60-70%) and oxygenated-aromatics (15-25%),
explained all AMS-determined SOA at low loads (thrust 3-5%) and most of the SOA formed (by 80%) at idle 6-
7% (Figure 5). Predicted aromatic SOA from benzene ($C_6H_6$), C2-benzenes ($C_8H_{10}$), C3-benzenes ($C_9H_{12}$), C4-
benzenes ($C_{10}H_{14}$), dimethylstyrenes ($C_{10}H_{12}$), toluene ($C_7H_8$), methylbenzaldehydes ($C_8H_8O$) and phenol
($C_6H_6O$) accounted for 60% of the AMS-determined SOA at 3-5% thrust (Figure 5). These results are consistent
with those previously obtained using a smog chamber, confirming that aromatic compounds are indeed
important SOA precursors in jet-engine emissions (Miracolo et al., 2011). Only a small fraction of these
compounds was determined in previous experiments using GC/MS measurements and therefore traditionally
considered as SOA precursors in models. Here, compared to previous experiments we show that non-traditional
aromatic and oxy-aromatic compounds, including naphthalene and its alkyl derivatives, C>3 alkyl derivatives of
single ring aromatics, and phenols, can explain the gap between measured SOA and SOA predicted based on
traditional precursors.
Exhaust-aging experiments were repeated 6 times at thrust 3-5% and the oxidation of NMOGs varied during
each of these aging experiments. Error bars shown in Figure 5 denote this variability in NMOG oxidation (in the
PAM) during aging experiments of the same thrust level. Indeed, errors related to yield values used may
significantly influence the results. These errors may be systematic and are complex to assess. They can be
affected by potential differences between the oxidation conditions in chambers and in the PAM (*e.g.* NOx, RH,
particle mass). Yields obtained with the PAM are consistent with those obtained from chambers (Bruns et al.,
2015), therefore we do not expect large systematic errors in the SOA predicted. However, based on the
variability of yields in previous chamber experiments we estimate the accuracy of our prediction to be within a
factor of 2, indicating that within our uncertainties a significant fraction of the precursors was identified.
NMOGs, including aromatic gases, were reduced with increasing thrust (from thrust 3-5% to thrust 6-7%) due
to more efficient operation of the turbine engine. This decrease amounted to 40% for the sum of aromatic HCs
and corresponded to a 30% decrease in SOA EI. Therefore, a more efficient engine operation implies less
NMOG emissions and reduced SOA formation potential at idle.

### 3.4 SOA formation at an approximated cruise load

A comparison of the predicted SOA with the SOA determined by the AMS is presented in Figure 6 at cruise
loads (top panel). Figure 6 also shows the SOA contribution predicted by the oxidation of NMOGs in the PAM
(bottom panel) under the same engine conditions. The SOA EI was 0.07 g/kg fuel for cruise load. The predicted
SOA fraction accounted for only 30% of the AMS-determined SOA (green bar, Figure 6) during cruise load
experiments. Aromatic SOA (predicted) accounted for only 4% of the AMS-determined SOA during these
experiments. The major fraction of the remaining SOA mass that was assigned to the identified precursors was
predicted to be from oxygenated NMOG molecules (Figure 6). Another 6% of the determined SOA may
originate from non-aromatic HCs (aliphatics and HC fragments > C6).
Predicted SOA was significantly lower compared to the measured SOA. While SOA precursors remain
unidentified under these conditions, several hypotheses might explain the observation. First, we could not
determine the contribution of alkanes smaller than 9 carbon atoms to the formed SOA, because these



compounds are not directly detected by the PTR-ToF-MS. Depending on the number of carbon atoms in their
molecular structure, the SOA potential of many alkanes may be comparable to that of single-ring aromatic
hydrocarbons (Tkacik et al., 2012) and therefore may play a role in the formation of the observed SOA.
However, our data do not suggest that a great part of the observed SOA is from non-measured alkanes, as we do
not observe any increase in the contribution of hydrocarbon fragments in the PTR-MS compared to idling
emissions. Second, the oxidation of primary semi-volatile compounds may yield significant SOA, because of
their elevated yields of near unity (Robinson et al., 2007). However, we note that these semi-volatile precursors
would play an important role at low aerosol concentrations, when most of these precursors reside in the gas-
phase where they can be oxidized. Under our conditions, concentrations range between 10 and 50 μg m$^{-3}$ and a
substantial fraction of these products resides already in the particle phase. Therefore the oxidation of these
products in the gas-phase by OH is unlikely to explain the observed entire 10-fold increase in the OA mass upon
oxidation, but only part of the mass. Finally, the PTR-MS data suggest that a great part of the precursors
measured are highly oxygenated gases, with O:C ratios ranging from 0.2 to 0.7, including, among others,
anhydrides (*e.g.* phthalic, succinic and maleic) and quinone derivatives. Unlike aromatic compounds and
alkanes present in the fuel, these compounds are likely formed at high temperature during combustion. The SOA
yields of these compounds remain unknown and it is likely that the yield value of 0.15 used here is a lower
estimate, which would result in an underestimation of the contribution of these compounds to the observed
SOA. We also note that unlike precursors detected under idle conditions, the ionization efficiency and the
fragmentation pattern of these compounds in the PTR-MS are highly uncertain, resulting in large uncertainties
in our predicted SOA. Therefore, results in Figure 6 should be considered with care. Notwithstanding these
uncertainties, we note that at cruise conditions the SOA contribution to the total secondary PM is minor
compared to sulfate and therefore these uncertainties have little impact on the implications of our results.
**4    Conclusions and implications for ambient air quality**
Gas-phase primary emissions and SA formation from an in-production turbofan were investigated in a test cell.
The engine loads (thrusts) during experiments were selected to simulate different aircraft operations. These
operations are summarized as landing take-off (LTO) cycle under four modes taxi/idle, approach, climb and
take-off with corresponding engine loads of 3-7%, 30%, 85% and 100%, respectively. In addition an
approximated cruising load (60%) was selected.
At idle conditions, SOA formation was mostly attributed to the oxidative processing of aromatic gases. Benzene
derivatives together with phenol were predicted as the major SOA precursors for an idling aircraft. Meanwhile,
during cruise load the emission of aromatic compounds was much lower and only explained a minor fraction of
SOA (4%). During these conditions, however, sulfate was found to dominate SA, contributing ~85% of the total
mass of aged aerosols and therefore its fraction is more relevant aloft.
The oxidation of NMOGs in the PAM yielded a SOA EI 100 times greater than POA under idling conditions
and 10 times greater at cruise load. According to our calculated production rates SOA from airport emissions
(idling jet engines) exceeds POA by a factor of 10 after only 3 hours of atmospheric aging and therefore
considerably impacts urban areas downwind of airport emissions. Compared to idling aircraft emissions, aging
of vehicle exhaust emissions results in much lower enhancements, ranging between factors of 5-10 and 1.5-3,
for gasoline and diesel vehicles, respectively (Gordon et al., 2014a; 2014b).
The NMOG emission factors and SOA potential can be used in conjunction with emission inventories and fuel
use data to assess the impact of aircraft emissions on air quality in comparison with other mobile sources. Here,
we have considered the Zurich international airport as an example (Switzerland, 23 million passengers in 2010).
Combining the recorded aircraft fuel use with the standard LTO cycle and the NMOG EIs measured, we
estimate aircraft NMOG emissions in Zurich for 2010 to be in the range of 90−190 tons/year (Kilic et al., 2017).
Based on the average SOA bulk yields (SOA/total NMOG) obtained herein (~5-8%), we estimate a total SOA
production potential from airport emissions for the area of Zurich to range from 5.4 to 13.2 tons/year. These
SOA production potential values can be directly compared to emissions from on-road vehicles derived from the
EDGARv4.2 emission inventory, which provides worldwide temporally and spatially resolved NMOG
emissions from road vehicles with a grid size of ~200 km$^2$. For the grid cell containing Zurich (47.25° North,
8.75° East) the NMOG emissions from on-road vehicles is estimated to be 631 tons/year. While SOA yields
from diesel vehicle emissions are expected to be more elevated than those from gasoline car emissions, due to
the presence of intermediate volatility species, recent reports suggest these yields to be comparably high, ~15%
(Gentner et al., 2017). Using this yield value for emissions from both types of vehicles (Platt et al., 2017), we
estimate the total SOA production potential from on road vehicles for the area of Zurich to be ~94 tons/year, 10
fold higher than SOA from aircraft emissions. However, the airport is a point source within this region and thus
the relative contribution of the airport emissions to a specific location downwind of this source is significantly
higher than implied by this calculation. Although this estimate applies to a specific airport, it does indicate that
aircraft NMOG emissions may constitute significant SOA precursors downwind of airports, while other fossil
fuel combustion sources dominate urban areas in general.





**Acknowledgements**
Funding was provided by the Swiss Federal Office of Civil Aviation (FOCA). This project would not have been
possible without the support of Rene Richter (PSI) and SR Technics personnel. Many individuals from SR
Technics contributed to the project but we owe special thanks to those from the Maintenance and Test Cell
Group. JGS acknowledges support from the Swiss National Science Foundation starting grant BSSGI0_155846.

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

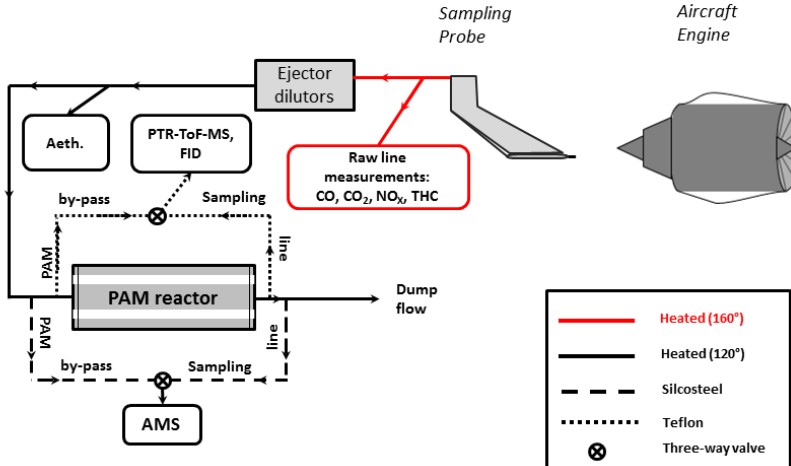

**Figure 1: Simplified scheme of the experimental setup.**





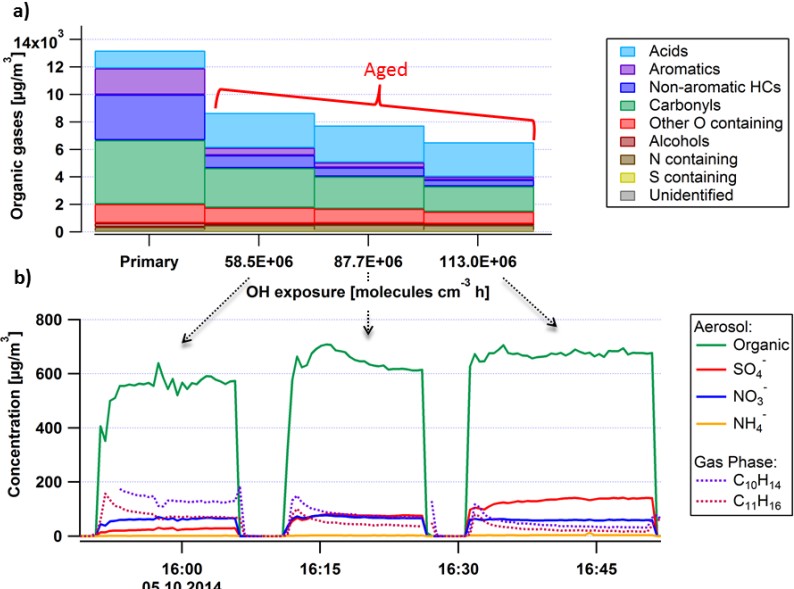

**Figure 2: Sample experiment showing mean NMOG emissions (top) and representative time series for particle and NMOG components (bottom) for varying OH exposures.** Hydrocarbon concentrations (non-aromatic HCs (dark blue), aromatic HCs (purple) and carbonyls (green) decrease in the PAM while the concentrations of acids (mostly formic and acetic ~90% of the total acids) increase. The bottom panel shows the aerosol (Organic, $SO_4$, $NO_3$, $NH_4$) formed and gaseous aromatics ($C_{10}H_{14}$, $C_{11}H_{16}$) for the different OH exposures in the PAM given in the top panel.

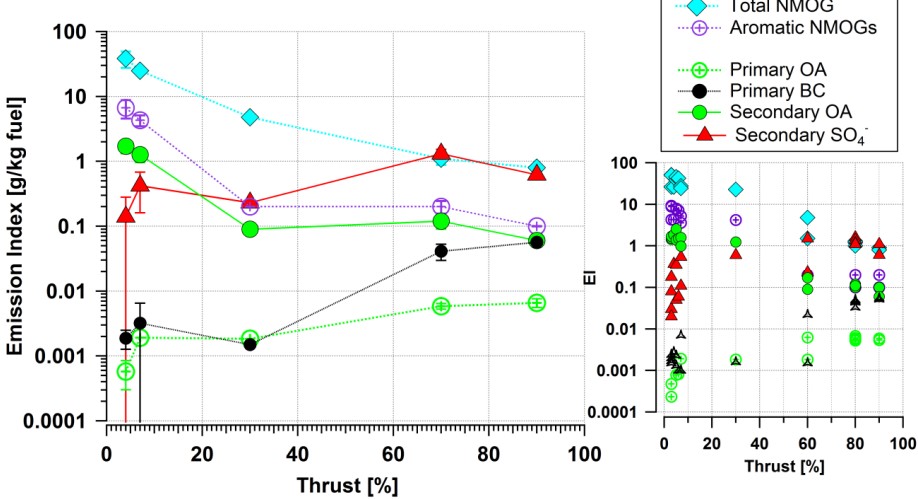

**Figure 3: Average emission indices (left) and EIs from individual test points (right)** for primary non-methane organic gases (NMOGs), aromatic gases, primary organic aerosol (POA), equivalent black carbon (BC), secondary organic aerosol (SOA), nitrate ($NO_3$) and sulfate ($SO_4$). Error bars (+/-) are the standard deviations of the means with a confidence interval of 95%.The OH exposure was in the range of 91-113 x $10^6$ molecule cm$^{-1}$ h for the secondary aerosol cases.





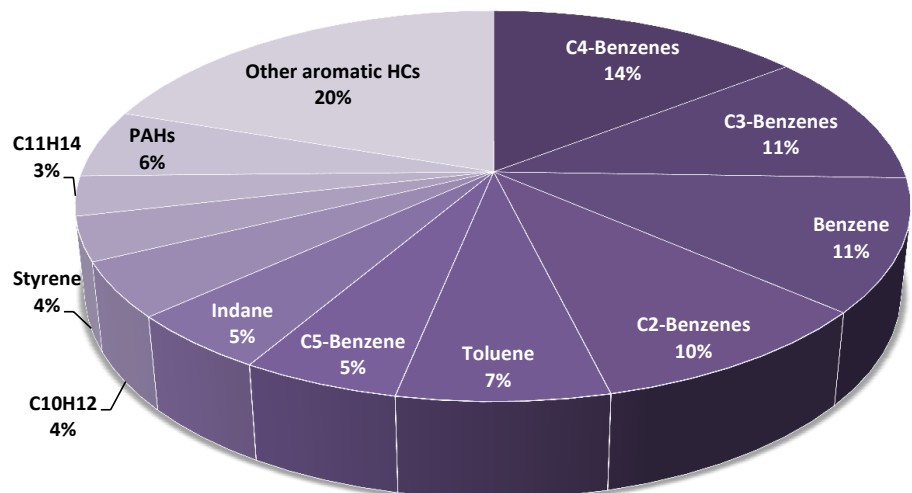

520
**Figure 4: Mass fractions of aromatic compounds for primary emissions (directly emitted) at idle
(thrust 3-7%).** Benzene derivatives, xylenes, tri-, tetra-, pentamethylbenzene, benzene and toluene
account for ~60% of all aromatics.

524



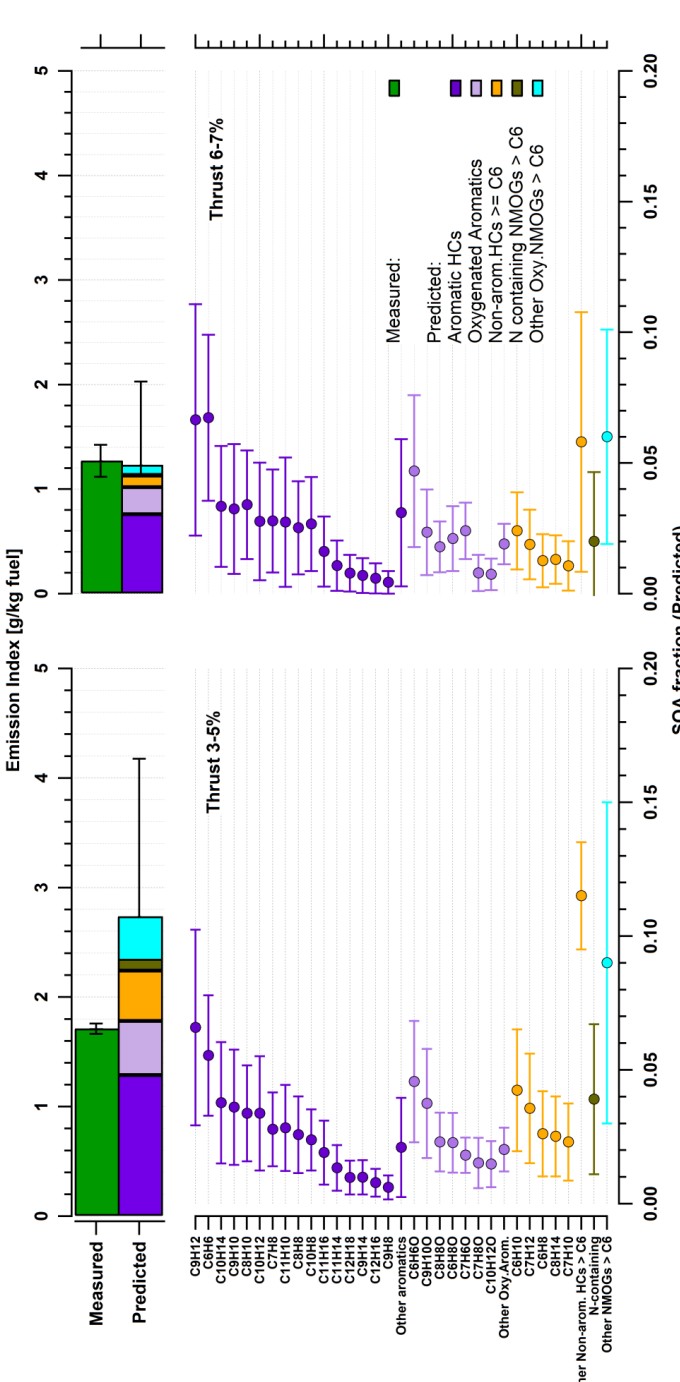

**Figure 5: Comparison of the SOA measured by the AMS and the SOA predicted by the oxidation of NMOGs (top panel).** The statistics are presented for low load idling (thrust 3-5%) on the left half and for idling 6-7% on the right. Aromatic hydrocarbons (purple) were the most abundant precursors of SOA at idle (thrust 3-7%) explaining all SOA formed (green, top panel) at thrust 3-5% and most (~90%) at thrust 6-7%. Aromatic SOA comprised the largest fraction followed by oxygenated aromatics (light purple - *e.g.* phenol, benzaldehydes), non-aromatic hydrocarbons (orange) with more than 6 carbon atoms in their molecular structure (non-arom. HCs ≥ C6), nitrogen containing compounds (brown), other oxygenated-NMOGs > C6 (cyan). Average fractions of individual NMOGs (bottom panel) were calculated by using SOA yields from literature (see *Table 2*) and the amount of NMOG reacted. Error bars show standard deviations of the means (CI: 95%).



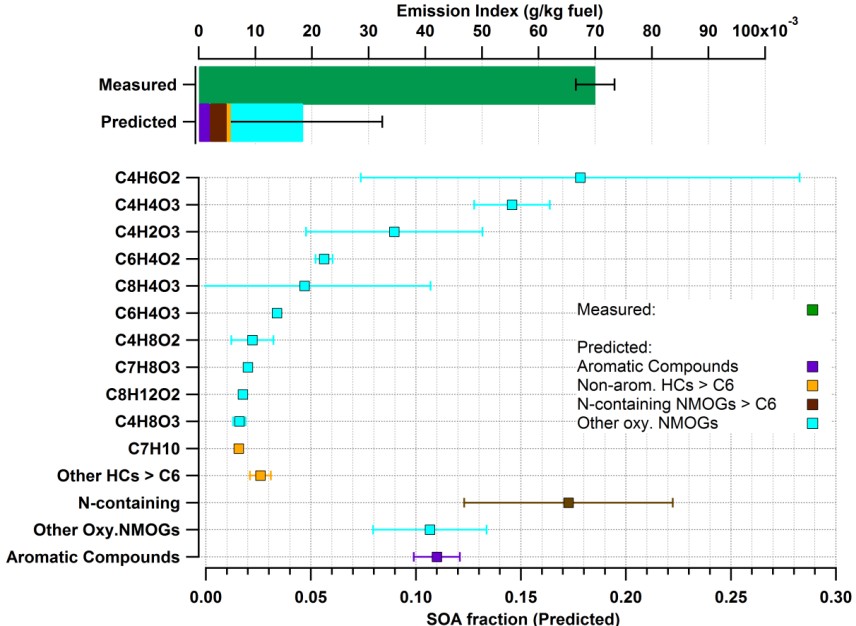

**Figure 6: Measured and predicted SOA comparison at an approximated cruise load,** using the same approach as in Fig. 5. In contrast to idle conditions, total NMOGs detected do not explain SOA formed.





**Table 1:** Volume mixing ratios of gaseous emissions, engine parameters and emission indices (EIs) for primary (directly emitted) and secondary (after aging) for all experiments.

| Thrust (%) | Fuel Consumption (kg/sec) | Primary | | | | | | | | Aged | | |
|---|---|---|---|---|---|---|---|---|---|---|---|---|
| | | $CO_2$ (ppmv) | CO (ppmv) | THC (ppmvC) | $NO_x$ (ppmv) | NMOG (g/kg fuel) | Aromatic Gases (g/kg fuel) | BC (g/kg fuel) | POA (g/kg fuel) | $SO_4$ (g/kg fuel) | OH exposure (molec. cm$^{-3}$ h) | SOA (g/kg fuel) |
| 3 | 0.09 | 1863 | 764 | 239 | 14 | 51 | 9.4 | 1.70E-06 | 2.34E-04 | 0.03 | 8.80E+07 | 1.4 |
| 3 | 0.09 | 1831 | 766 | 244 | 14 | 51 | 8.8 | 2.00E-06 | 4.74E-04 | 0.02 | 8.80E+07 | n/a |
| 3 | 0.09 | 1560 | 709 | 222 | 14 | 26 | 4.3 | 1.50E-06 | <2.0E-03 | 0.08 | 6.00E+07 | 1.5 |
| 3 | 0.09 | 1560 | 709 | 222 | 14 | 26 | 4.3 | 2.40E-06 | <2.0E-03 | 0.18 | 9.00E+07 | 1.7 |
| 3 | 0.09 | 1560 | 709 | 222 | 14 | 26 | 4.3 | 2.80E-06 | <2.0E-03 | 0.37 | 1.13E+08 | 1.9 |
| 4 | n/a | 1884 | 543 | 173 | n/a | 39 | 7.7 | 2.30E-06 | <2.0E-03 | 0.35 | 1.00E+08 | 2.6 |
| 5 | 0.11 | 1829 | 442 | 114 | 17 | 46 | 7.9 | 1.30E-06 | 7.81E-04 | 0.05 | 8.80E+07 | 1.4 |
| 5 | 0.11 | 1758 | 422 | 98 | 17 | 42 | 7.1 | 1.00E-06 | 8.13E-04 | 0.06 | 8.80E+07 | 1.5 |
| 6 | 0.12 | 1934 | 410 | 113 | 21 | 28 | 5.2 | 1.00E-06 | <2.0E-03 | 0.54 | 1.00E+08 | 1.6 |
| 7 | 0.14 | 1909 | 168 | 45 | 23 | 25 | 3.6 | 7.00E-06 | 1.91E-03 | 0.11 | 8.80E+07 | 1.0 |
| 7 | 0.14 | 1978 | 385 | 94 | 23 | 23 | 4.2 | 1.60E-06 | <2.0E-03 | 0.6 | 1.00E+08 | 1.2 |
| 30 | 0.31 | 2953 | 40 | 8 | 62 | 5 | 0.2 | 1.50E-06 | 1.84E-03 | 0.23 | 8.80E+07 | 0.1 |
| 60 | 0.65 | 3709 | 12 | 7 | 131 | 2 | 0.2 | 2.20E-05 | 6.21E-03 | 1.51 | 8.50E+07 | 0.2 |
| 60 | 0.65 | 3709 | 12 | 7 | 131 | 1 | 0.2 | 3.30E-05 | 6.80E-03 | 1.61 | 8.50E+07 | 0.1 |
| 80 | 0.9 | 4344 | 16 | 6 | 195 | 1 | 0.1 | 4.60E-05 | 5.55E-03 | 1.39 | 1.00E+08 | 0.1 |
| 80 | 0.9 | 4291 | 14 | 7 | 195 | 1 | 0.2 | 4.40E-05 | 6.01E-03 | 1.16 | 8.50E+07 | 0.1 |
| 80 | 0.9 | 4291 | 14 | 7 | 195 | 1 | 0.2 | 5.00E-05 | 5.24E-03 | 1.1 | 8.50E+07 | 0.1 |
| 80 | 0.9 | 4291 | 14 | 7 | 195 | 1 | 0.2 | 5.20E-05 | 5.37E-03 | 1.1 | 8.50E+07 | 0.1 |
| 90 | 1.1 | 4657 | 16 | 6 | 202 | 1 | 0.1 | 5.50E-05 | 5.89E-03 | 0.61 | 8.50E+07 | 0.1 |
| 90 | 1.1 | 4657 | 16 | 6 | 202 | 1 | 0.1 | 5.80E-05 | 7.30E-03 | 0.63 | 8.50E+07 | 0.1 |





**Table 2. Precursor OGs, their corresponding functional group and protonated m/z, SOA yield coefficients\*** from literature, and average SOA EI estimated for different thrusts.

| VOC formula | m/z | Group | SOA yield* | SOA Emission Index (g/kg fuel) | | | |
| | | | | Thrust 3-5% | | Thrust 6-7% | |
| | | | | Average | ± | Average | ± |
|---|---|---|---|---|---|---|---|
| $(C_9H_{12})H^+$ | 121.101 | Aromatic | 0.32 | 0.19 | 0.11 | 0.07 | 0.04 |
| $(C_6H_6)H^+$ | 79.054 | Aromatic | 0.33 | 0.16 | 0.06 | 0.07 | 0.03 |
| $(C_{10}H_{14})H^+$ | 135.117 | Aromatic | 0.2 | 0.11 | 0.07 | 0.03 | 0.02 |
| $(C_9H_{10})H^+$ | 119.086 | Aromatic | 0.32 | 0.10 | 0.06 | 0.03 | 0.02 |
| $(C_8H_{10})H^+$ | 107.086 | Aromatic | 0.2 | 0.10 | 0.05 | 0.04 | 0.02 |
| $(C_{10}H_{12})H^+$ | 133.101 | Aromatic | 0.32 | 0.10 | 0.06 | 0.03 | 0.02 |
| $(C_7H_8)H^+$ | 93.070 | Aromatic | 0.24 | 0.08 | 0.04 | 0.03 | 0.02 |
| $(C_{11}H_{10})H^+$ | 143.086 | Aromatic | 0.52 | 0.08 | 0.05 | 0.03 | 0.02 |
| $(C_8H_8)H^+$ | 105.070 | Aromatic | 0.32 | 0.07 | 0.04 | 0.03 | 0.02 |
| $(C_{10}H_8)H^+$ | 129.070 | Aromatic | 0.52 | 0.07 | 0.03 | 0.03 | 0.02 |
| $(C_{11}H_{16})H^+$ | 149.132 | Aromatic | 0.2 | 0.05 | 0.03 | 0.02 | 0.01 |
| $(C_{11}H_{14})H^+$ | 147.117 | Aromatic | 0.2 | 0.04 | 0.02 | 0.01 | 0.01 |
| $(C_{12}H_{18})H^+$ | 163.148 | Aromatic | 0.2 | 0.03 | 0.02 | 0.01 | 0.01 |
| $(C_9H_{14})H^+$ | 123.117 | Aromatic | 0.2 | 0.03 | 0.02 | 0.01 | 0.01 |
| $(C_{12}H_{16})H^+$ | 161.132 | Aromatic | 0.2 | 0.02 | 0.01 | 0.01 | 0.01 |
| $(C_9H_8)H^+$ | 117.070 | Aromatic | 0.2 | 0.02 | 0.01 | 0.00 | 0.00 |
| Other | - | Aromatic | 0.2 | 0.06 | 0.05 | 0.00 | 0.00 |
| $(C_6H_6O)H^+$ | 95.049 | Oxy-arom | 0.44 | 0.13 | 0.07 | 0.05 | 0.03 |
| $(C_6H_8O)H^+$ | 97.065 | Oxy-arom | 0.32 | 0.07 | 0.03 | 0.02 | 0.01 |
| $(C_7H_6O)H^+$ | 107.049 | Oxy-arom | 0.32 | 0.05 | 0.02 | 0.02 | 0.01 |
| $(C_6H_6O_2)H^+$ | 111.044 | Oxy-arom | 0.39 | 0.03 | 0.01 | 0.01 | 0.00 |
| $(C_{10}H_{12}O_2)H^+$ | 165.091 | Oxy-arom | 0.2 | 0.00 | 0.00 | 0.00 | 0.00 |
| $(C_{10}H_{14}O_2)H^+$ | 167.107 | Oxy-arom | 0.2 | 0.00 | 0.00 | 0.00 | 0.00 |
| Other NMOGs | > 79.054 | Other NMOG | 0.15 | 1.15 | 0.74 | 0.20 | 0.02 |

\*SOA yields from (Ng et al., 2007; Alvarez et al., 2009; Chan et al., 2009 ; 2010; Hildebrandt et al., 2009; Shakya and Griffin, 2010; Chhabra et al., 2011; Nakao et al., 2011; Yee et al., 2013)