# Peer review of "Identification of secondary aerosol precursors emitted by an aircraft turbofan"

_Atmospheric Chemistry and Physics, 2017_

## Referee Comment (RC1) · Anonymous Referee #1 · 7 Dec 2017

This is a very will written and informative paper on a quite relevant and timely subject regarding aircraft PM emissions. The authors have extended work in this area in important ways, with proper and appropriate recognition of the prior work that they are building upon. The results are interesting, important, and appropriately qualified for the application of these data in assessing the environmental impact of aviation emissions. Their brief discussion of impacts in the conclusions section will help consumers of this data to use it for impact assessments.

This paper is very well written, and I noted no typographical errors or mistakes.

My only comment is the following.

In the discussion of Figure 6, the authors note that the measured SOA at "cruise" is

significantly higher than that predicted based on oxidation and gas-to-particle conversion of the measured precursors. They go on to point out that the levels are quite low at "cruise", so that this doesn't affect their primary conclusions. I understand and do not take issue with any of those comments.

My suggestion to the authors is to consider a contribution to the measured SOA from engine oil. Much work has been done to quantify the engine oil contribution to the volatile PM (Timko et al., 2010, Yu et al., 2010, Timko et al., 2014 and others), where the mass spectra of oil compounds from AMS data were identified and analyzed. An oil contribution would add to the measured SOA, yet not be a factor in the measured gas phase emissions. Thus it might help explain the lack of mass closure in the budget in Figure 6. The engine oil contribution is not combustion-related, and thus will not have the significant power variation as seen the the gaseous combustion emissions precursors. As such, the oil contribution might not change as much with power. This means that it could be a significant component at "cruise" power conditions where SOA is low, yet be negligible at idle where the gaseous emissions are much higher.

I do not know whether the authors can identify oil fragment peaks in the AMS data, but I would suggest they look. At the very least, I would suggest that in the discussion for Figure 6, they acknowledge that engine oil is a possible explanation for their lack of mass closure on the SOA budget.

---

## Referee Comment (RC2) · Anonymous Referee #2 · 3 Jan 2018

The manuscript presents data from an airport test cell, where emissions from an engine are aged in a chamber by exposure to OH and the predicted vs measured SOA are compared for several thrust conditions. These measurements are then used to assess the potential impact on LAQ with an airport as a point source.

The paper is well written and very timely with increased interest in PM from aircraft and the upcoming nvPM regulation. There are a few questions/queries listed below. My only main suggestion is that the abstract/conclusion should reflect the fact only one of the three modelling cases matches the measured SOA. In the low thrust case, the model over predicted the SOA mass by a factor of $\sim$1.6 and (as stated) at cruise it under predicts by a factor $\sim$0.25. Lines 21-22, to say that the SOA mass can be explained by the oxidation of gaseous aromatic species is only true for 1 case. The

body of the text does however consider the contributions of low Cn products or yields not being correct.

1) In most chamber studies, it is usual to consider the wall losses and the calculated mass is derived from the difference between the measured mass and some empirical loss model. Is this something that has been considered here?

2) Did the authors check the sulphate fragmentation? Either by Hi Res analysis or the 48:64 etc ratios? Are they sure it is not an organic fragment interfering with a sulphate peak? Same for the nitrate.

3) Are the authors sure that the measured particles are within the AMS 100% transmission window ($\sim$60-600nm Dva) for all thrust setting and the CE $\sim$ 1? Line 147 implies the AMS mass was converted to a volume and compared with the SMPS volume. Is that correct? Are the authors assuming that under organic rich conditions, the contribution to the total volume from the eBC is negligible or that the OA is externally mixed from the eBC? Is this the case across all thrust settings? Furthermore, what shape factor did the authors use to convert the Dva to Dm or vice versa to compare the total volumes? Or did the authors use the PToF data and an effective density, in which case is one value across all powers sensible?

4) Section 3.3 – Can the authors expand on why there is a significant difference between the predicted and measured SOA at 3-5% thrust, compared with almost complete agreement at 6-7% thrust? They allude to a reduced NMOG contribution at 6-7% thrust, could it be that not all the NMOG is contributing to the mass? Could the difference in NOx explain this? Other chamber studies show different chemical pathways based on the NOx concentrations. Elsewhere in the manuscript, the authors discount the possibility of oxidation of the semi-volatiles because they are already bound in the particle phase. With the dilutors, is it possible that has been a re-partitioning due to the dilution of high purity N2 which will change the equilibrium between particle and the gas?

5) Given the work of Yu (and others referenced in the manuscript), can the contribution from the oil to the SOA be determined? Are the authors looking at primarily combustion processed OA or oil processed OA?

6) I am assuming the AMS used was a HR-ToF-AMS as they authors refer to V-mode? Has no Hi-Res (PIKA) analysis been done on the data? This may give insight into some of the oxidation products (although may be beyond the scope of the paper).

7) The work on the estimated yearly production of SOA from Zurich is based on the reported NMOG EIs and the work of the authors, but the PAM is under photochemical oxidation conditions i.e. OH production during daylight hours. Do the authors estimation take into account daytime vs nightime activity on the emissions of NMOG and the potential impact on SOA formation?

Minor typos/suggestions:

Lines 192 & 195. NRPM1 or NR-PM1. Check document is consistent. Same with eBC or BC. Both are used in the document.

Figure 2: Labelled a) and b) but then not referenced in the caption as a) and b). Either have a & b in the caption or remove from figure.

In other figures where there are 2 graphs, authors do not label them a) or b). Have a consistent format for the figures.

---

## Author Comment (AC1) · 1 Mar 2018

**Response to Reviews**

Manuscript acp-2017-907

**Anonymous Referee #1**

We thank the referee for a thorough review of our manuscript. The referee's comment on lubricant oil contribution to the organic PM was very valuable and we believe that addressing this issue considerably improves the manuscript. Please find in the following a point-by-point response (in regular typeset, black font) to the comments (in italic typeset, blue font). Changes in the manuscript are indicated in italic typeset, grey font.

In the discussion of Figure 6, the authors note that the measured SOA at "cruise" is significantly higher than that predicted based on oxidation and gas-to-particle conversion of the measured precursors. They go on to point out that the levels are quite low at "cruise", so that this doesn't affect their primary conclusions. I understand and do not take issue with any of those comments.

My suggestion to the authors is to consider a contribution to the measured SOA fromengine oil. Much work has been done to quantify the engine oil contribution to the volatile PM (Timko et al., 2010, Yu et al., 2010, Timko et al., 2014 and others), where the mass spectra of oil compounds from AMS data were identified and analyzed. An oil contribution would add to the measured SOA, yet not be a factor in the measured gas phase emissions. Thus it might help explain the lack of mass closure in the budget in Figure 6. The engine oil contribution is not combustion-related, and thus will not have the significant power variation as seen the the gaseous combustion emissions precursors. As such, the oil contribution might not change as much with power. This means that it could be a significant component at "cruise" power conditions where SOA is low, yet be negligible at idle where the gaseous emissions are much higher.

I do not know whether the authors can identify oil fragment peaks in the AMS data, but I would suggest they look. At the very least, I would suggest that in the discussion for Figure 6, they acknowledge that engine oil is a possible explanation for their lack of mass closure on the SOA budget.

From a sampling point of view, there are technical differences on assessing the lubricant oil contribution to the exhaust in our sample and in a sample collected on the runway because our measurements were conducted on the engine core flow only. The turbine engine exhaust was sampled by a single-point probe with an inner diameter of 8 mm, located 0.7 m downstream of the engine exit plane. The lubricant oil for the engine model studied is emitted at a different location through the so called "engine center body (like an oil tube)". Therefore, the engine studied here does not vent lube oil through its core. Our sampling location does not and cannot capture emissions from this center body. There might be still a minimal amount of lube oil passing through the engine core due to leaking hydrodynamic seals (most likely at startup and idle). However, the PM fraction originating from lubricant oil is not comparable to what is seen

**on the runway. We added following comments to the original manuscript (see experimental setup section-lines 76-79) to better define sampling conditions:**

At this sampling location, the lubricant oil contribution to the exhaust is expected to be minimal (only due to leaking hydrodynamic seals at startup/idle) compared to the runway measurements since the engine design studied does not vent lubricant oil through its core (where probe pulled sample).

Lubricant oil contribution to the PM could be a potentially significant source for primary particulate emissions especially at cruise loads. After reviewing the previous studies on quantifying oil-sourced PM in aircraft exhaust, these also include studies suggested by Referee 1, we now also point out this missing source in PM emissions in the original manuscript (lines 230-233):

There is a PM fraction in the engine exhaust originating from the lubricant oil (Yu et al., 2012) that is not measured here due to the sampling location and engine model studied. The EI of this PM fraction ranges in 2 - 10 mg/kg fuel (Yu et al, 2010) for other engine models than the engine model studied here. The PM originating from lubricant oil is less than 1% of the SOA at idle. However it may be a significant source at cruise loads.

In addition, we included fuel specifications with a supplement which we think as important to know about the sources of gaseous organics originating from the fuel itself.

| Parameter              | Unit    | Annex 16
LOW | Annex
HIGH | : 16
H | pre
campaign | during
campaign | post
campaign |
|------------------------|---------|-----------------|---------------|-----------|-----------------|--------------------|------------------|
| Aromatics              | % (V/V) | 15              |               | 23        | 18.3            | 17.9               | 17.8             |
|                        | %       |                 |               |           |                 |                    |                  |
| Sulfur, total          | (m/m)   | 0               |               | 0.3       | 0.05            | 0.046              | 0.044            |
| Initial boiling point  | °C      | NA              | NA            |           | 151             | 151                | 151              |
| 10 Vol % recovered at  | °C      | 155             |               | 201       | 167             | 166                | 167              |
| 20 Vol % recovered at  | °C      | NA              | NA            |           | 173             | 172                | 172              |
| 50 Vol % recovered at  | °C      | NA              | NA            |           | 193             | 192                | 192              |
| 90 Vol % recovered at  | °C      | NA              | NA            |           | 237             | 235                | 236              |
| End point              | °C      | 235             |               | 285       | 262             | 259                | 259              |
| Residue                | % (V/V) | NA              | NA            |           | 1.1             | 1                  | 1                |
| Loss                   | % (V/V) | NA              | NA            |           | 0.6             | 0.6                | 0.5              |
| Density at 15 °C       | kg/m³   | 780             |               | 820       | 798.4           | 798.9              | 799              |
| Viscosity at -20 °C    | mm²/s   | 2.5             |               | 6.5       | 3.571           | 3.55               | 3.567            |
| Specific energy, net   | MJ/kg   | 42.86           | 4             | 13.5      | 43.2            | 43.2               | 43.2             |
| Smoke point            | mm      | 20              |               | 28        | 23              | 24                 | 23               |
| Naphthalenes           | % (V/V) | C               |               | 3.5       | 0.81            | 0.80               | 0.76             |
|                        | %       |                 |               |           |                 |                    |                  |
| Hydrogen               | (m/m)   | 13.4            | 1             | L4.3      | 14.2            | 14.2               | 14.4             |
| H/C ratio (calculated) | NA      | 1.84            | 1             | L.99      | 1.97            | 1.97               | 2.00             |
| Origin                 |         |                 |               |           | Test Cell       | Test Cell          | Test Cell        |
| Campaign               |         |                 |               |           | A-Pride 7       | A-Pride 7          | A-Pride 7        |

**Table S1 Fuel Specifications (ASTM fuel parameters)**

**Anonymous Referee #2**

We thank you for the constructive comments that will significantly enhance the quality of the manuscript. Please find in the following a point-by-point response (in regular typeset, black font) to the comments (in italic typeset, blue font). Changes in the manuscript are indicated in italic typeset, grey font.

The manuscript presents data from an airport test cell, where emissions from an engine are aged in a chamber by exposure to OH and the predicted vs measured SOA are compared for several thrust conditions. These measurements are then used to assess the potential impact on LAQ with an airport as a point source.

The paper is well written and very timely with increased interest in PM from aircraft and the upcoming nvPM regulation. There are a few questions/queries listed below. My only main suggestion is that the abstract/conclusion should reflect the fact only one of the three modelling cases matches the measured SOA. In the low thrust case, the model over predicted the SOA mass by a factor of ~1.6 and (as stated) at cruise it under predicts by a factor ~0.25. Lines 21-22, to say that the SOA mass can be explained by the oxidation of gaseous aromatic species is only true for 1 case. The body of the text does however consider the contributions of low Cn products or yields not being correct.

We would like to point out that SOA closure from gas-phase reactivity is often very challenging. Here, we use literature yields obtained during chamber studies to estimate the amount of SOA formed in the oxidation flow reactor, PAM. These yields are very sensitive to the conditions at which the experiments were conducted. Parameters that can have an influence of SOA yield determination include among others the NOX/VOC ratio and the particle condensational sink and mass concentrations. Under our conditions, the PAM operated under low NOX/VOC conditions and at rather high condensational sink; both conditions favor higher yields. We have attempted to use yields from studies conducted under similar conditions. However, we note that reported yields for similar conditions may vary by up to a factor of two. Therefore, we do not expect that we can achieve closure better than a factor of two. The left panel of Figure 5 shows that the estimated SOA emission index is 1.6 times higher than the measured SOA index, as indicated by the reviewer; however, this difference is not statistically significant within the measurement variability. Therefore, we considered that for this case the model actually reproduced the measurement adequately. By contrast, as mentioned in the manuscript, SOA emission indices estimated based on the measured precursors underestimates the measured SOA emission indices by a factor of four. This underestimation was systematic and therefore we consider that the model does not reproduce the data adequately and that other factors may play a role in SOA formation under these conditions. We clarify these points in the manuscript (lines 248 - 253):

SOA yields are sensitive to the conditions at which the experiments were conducted. Parameters that can have an influence on SOA yield determination include among others the  $NO_X/VOC$  ratio and the particle condensational sink and mass concentrations. Under our conditions, the PAM operated under low  $NO_X/VOC$  conditions and at rather high condensational sinks; both conditions would favor higher yields (Stirnweis et al. 2017). We have attempted to use yields from studies conducted under similar conditions. However, we note that reported yields for similar conditions may vary by up to a factor of two.

1) In most chamber studies, it is usual to consider the wall losses and the calculated mass is derived from the difference between the measured mass and some empirical loss model. Is this something that has been considered here?

We have considered particle losses to the PAM wall by measuring primary emissions of NR-PM1, with lights off, before and after the PAM. From this test, we estimated the losses to the PAM walls to be ~5%, consistent with previous studies (Bruns et al. 2015, Palm et al. 2016). All data presented were corrected for particle wall losses. This is mentioned in the new version of the manuscript (lines 187 - 190):

Particle losses to the PAM wall were taken into account during particle mass calculations by measuring primary emissions of NR-PM1, with lights off, before and after the PAM. From this test, we estimated the losses to the PAM walls to be ~5%, consistent with previous studies (Bruns et al. 2015, Palm et al. 2016). All data presented were corrected for particle wall losses.

**2) Did the authors check the sulphate fragmentation? Either by Hi Res analysis or the 48:64 etc ratios? Are they sure it is not an organic fragment interfering with a sulphate peak? Same for the nitrate.**

Despite that data presented in the manuscript are obtained using unit mass resolution, we have thoroughly checked for organic interference on the sulfate measurements, using high resolution data analysis. This interference is negligible for sulfate: the signals at m/z related to sulfate can be fully be attributed to sulfate aerosols. An example is shown below in Figure R1 for the fragment at m/z64. For nitrate, a non-negligible fraction can be attributed to the organic aerosol fragmentation (~25%). However, we do not report nitrate emission factors in the manuscript, because of other reasons: (1) nitrate is a negligible fraction compared to the organic aerosol (OA) and sulfate (

**Figure R1:** High resolution signal at m/z64, dominated by  $SO_2^+$  from the fragmentation of sulfate. The upper panel shows the high resolution fit residuals. The second, third and fourth panels show the signals when the AMS chopper was opened, closed (aerosol background) and the difference between opened and closed, respectively. The signal in Figure R1 is integrate for a period where the concentration of sulfate is 3.5 times lower than the OA concentrations.

3) Are the authors sure that the measured particles are within the AMS 100% transmission window ( $\sim$ 60-600nm Dva) for all thrust setting and the CE  $\sim$ 1? Line 147 implies the AMS mass was converted to a volume and compared with the SMPS volume. Is that correct? Are the authors assuming that under organic rich conditions, the contribution to the total volume from the eBC is negligible or that the OA is externally mixed from the eBC? Is this the case across all thrust settings? Furthermore, what shape factor did the authors use to convert the Dva to Dm or vice versa to compare the total volumes? Or did the authors use the PToF data and an effective density, in which case is one value across all powers sensible?

The reviewer has raised a number of points in this comment. We have broken the comment into specific points, addressing each one in turn.

Aerosol size distribution and AMS transmission window. The aerosol size distributions after the PAM are fairly consistent, with a small shift towards higher sizes at lower thrust levels, most likely due to enhanced condensation of oxidized organic vapors produced at low thrust. The bulk of the aerosol mass for the aged emissions is within the AMS transmission window (60-500nm), as can be seen in Figure R2 for different thrust levels. We estimate that particle losses in the AMS inlet for aged emissions are less than 15%. We note that the fraction of black carbon is clearly negligible in aged emissions (

**Figure R2:** Integrated particle mass distributions normalized by the total particle mass for low thrust (<9%) and high thrust (>10%), obtained from SMPS data. The Shaded areas indicate estimated lens cutoffs for the AMS, at smaller and larger particle sizes.

AMS collection efficiency due to particle bounce. AMS measurements typically suffer from low collection efficiency (CE) due to particle bounce on the vaporizer, which depends on the particle chemical composition. CE is significantly reduced in the presence of solid ammonium sulfate particles. For aged emissions, we have estimated the CE by comparing the particle mass calculated by the AMS + BC with that estimated using the SMPS. For both thrust settings, the CE is not significantly different than 1. This is because at low thrust the aged particles consist mostly of organic matter, which had been shown to be efficiently collected on the vaporizer for chamber aerosols (Stirnweis et al., 2017, Platt et al., 2017). Meanwhile, the aged aerosol at higher thrust settings is predominantly composed of sulfuric acid; the molar ratio between NH3 and SO4 is ~0.3, while fully neutralized ammonium sulfate would have a ratio of 2. As opposed to ammonium sulfate, sulfuric acid is efficiently collect on the vaporizer. Therefore, we concluded that using CE=1 for all thrust levels is adequate. We note that because of the high contribution of the BC to fresh emissions, we could not determine the CE of primary OA and we used CE=1. Such CE is expected for hydrocarbon-like particles.

We conclude that particle transmission and collection efficiency are well constrained for aged aerosols and have little influence on our results. However, primary emission factors are more

**uncertain and should be indeed considered as low estimates. We added this discussion in the corrected version of the manuscript (lines 149-157).**

For aged emissions, the CE was estimated by comparing the particle mass calculated by the AMS + BC with that estimated using the SMPS. For both thrust settings, the CE is not significantly different than 1. This is because at low thrust the aged particles consist mostly of organic matter, which had been shown to be efficiently collected on the vaporizer for chamber aerosols (Stirnweis et al., 2017, Platt et al., 2017). Meanwhile, the aged aerosol at higher thrust settings is predominantly composed of sulfuric acid; the molar ratio between NH3 and  $SO_4$  is ~0.3, while fully neutralized ammonium sulfate would have a ratio of 2. As opposed to ammonium sulfate, sulfuric acid is efficiently collect on the vaporizer. Therefore, we concluded that using CE=1 for all thrust levels is adequate. We note that because of the high contribution of the BC to fresh emissions, we could not determine the CE of primary OA and we used CE=1. Such CE is expected for hydrocarbon like particles.

4) Section 3.3 - Can the authors expand on why there is a significant difference between the predicted and measured SOA at 3-5% thrust, compared with almost complete agreement at 6-7% thrust? They allude to a reduced NMOG contribution at 6-7% thrust, could it be that not all the NMOG is contributing to the mass? Could the difference in NOx explain this? Other chamber studies show different chemical pathways based on the NOx concentrations. Elsewhere in the manuscript, the authors discount the possibility of oxidation of the semi-volatiles because they are already bound in the particle phase. With the dilutors, is it possible that has been a re-partitioning due to the dilution of high purity  $N_2$  which will change the equilibrium between particle and the gas?

Above, we have addressed part of this comment related to the challenges associated with achieving SOA closure from gas-phase reactivity. We have used literature yields obtained during chamber studies to estimate the amount of SOA formed in the PAM. These yields are very sensitive to the conditions at which the experiments were conducted and we have attempted to use yields from studies conducted under similar conditions. However, we note that reported yields for similar conditions may vary by up to a factor of two. Therefore, we do not expect that we can achieve closure within much less than a factor of two. At 3-5% thrust, the calculated SOA emission index is 1.6 times higher than the measured SOA index, well within a factor 2. In addition, this difference is not statistically significant within the measurement variability of the amount of SOA estimated from the gas-phase. Therefore, we think that the calculations capture adequately the measurements.

In the PAM, we can exclude the effect of NO on the  $RO_2$  chemistry and the yields, because of the very high  $O_3$  concentrations, which results in the titration of NO into  $NO_2$ . Considering the oxidation of additional primary semi-volatile species would result in further overestimation. Despite the dilution of the emissions, OA levels in the PAM were around 100 µg m-3 and under these conditions most of the semi-volatile species would reside in the particle phase. Therefore, we do not expect a high contribution of primary semi-volatile vapors oxidation products to the formed SOA.

**5) Given the work of Yu (and others referenced in the manuscript), can the contribution from the oil to the SOA be determined? Are the authors looking at primarily combustion processed OA or oil processed OA?**

The lubricant oil could indeed be a significant source of PM formed in the engine exhaust. However, we sample from the engine core by the probe located 0.7 m downstream of the engine exit, where the oil contribution is expected to be minimal (only due to a probable leak from hydrodynamic seals). Therefore, our SOA analyses focus on primarily combustion processed OA. Further details on the issue can be found in reply to Referee 1 for a similar question and following lines were added to updated manuscript to clarify:

"At this sampling location, the lubricant oil contribution to the exhaust is expected to be minimal (only due to leaking hydrodynamic seals at startup/idle) compared to the runway measurements since the engine design studied does not vent lubricant oil through its core (where the probe pulled sample)" (lines 77-80)

"There is a PM fraction in the engine exhaust originating from the lubricant oil (Yu et al., 2012) that is not measured due to the sampling location and engine model studied. The EI of this PM fraction ranges in 2 - 10 mg/kg fuel (Yu et al, 2010) for other engine models than the engine model studied here. This PM originating from lubricant oil is less than 1% of the SOA at idle however it could be a significant source at cruise loads." (Lines 220-223)

**6) I am assuming the AMS used was a HR-ToF-AMS as they authors refer to V-mode? Has no Hi-Res (PIKA) analysis been done on the data? This may give insight into some of the oxidation products (although may be beyond the scope of the paper).**

As mentioned above, we have indeed performed high resolution analysis on part of the data, mainly as a quality check. However, we note that the mass spectra recorded for aged OA are very similar (R>0.98). Most of the mass spectra indicate a high aerosol oxygen content, with the fraction of  $m/zCO_2^+ \sim 25\%$  and an O:C ratio  $\sim 2$ . This similarity between the mass spectra is due to the high OH exposure in the PAM resulting in the extensive oxidation of the emissions into chemically similar composition and the use of electron ionization by the AMS, which results in substantial fragmentation of the oxidized compounds. Therefore, the SOA mass spectra can provide only limited information on the precursors involved in SOA formation. Therefore, we have focused the paper on the absolute secondary particle emission indexes and the most important precursors. In future work, detailed molecular analysis, for example using chemical ionization techniques, could indeed help in providing more information on the oxidation products in the particle phase.

7) The work on the estimated yearly production of SOA from Zurich is based on the reported NMOG EIs and the work of the authors, but the PAM is under photochemical oxidation conditions i.e. OH production during daylight hours. Do the authors estimation take into account daytime vs nighttime activity on the emissions of NMOG and the potential impact on SOA formation?

For estimating the impact of airport emissions on SOA production, we have used the total annual emissions from Zurich airport and assumed that all of these emissions will be transformed into SOA. This consideration would indeed yield highest estimates, as only part of the emissions will be oxidized in the proximity of the airport (e.g. Zurich area) and oxidation will be diminished during the night at low OH concentrations. However, we have shown that most of the SOA mass is formed at moderate OH exposures, which signifies that SOA can be formed locally. In addition, we note that the flight activity is highest during summer and during day-time. The number of the day-time flights account for more than 95% of the daily flights, when highest amount of sunlight enables photochemical oxidation.

Additionally, a question on the variability of the NMOG EIs came up during the discussion with the co-authors, we added part of this discussion to clarify our assumptions while estimating SOA production (lines 335 - 338):

We note that the influence of the engine type and age on the total NMOG emission rates is minor compared to other parameters (e.g. thrust level). In addition, the engine type tested here is the most frequently used/sold model for commercial aviation. Thus, we believe that the emission rates used here are generally representative of idling aircraft emissions.

**Minor typos/suggestions:**

*Lines 192 & 195. NRPM1 or NR-PM1. Check document is consistent. Same with eBC or BC. Both are used in the document.*

"NRPM1" in the corresponding line was corrected as "NR-PM1" and the abbreviation of equivalent black carbon is now "BC" in entire manuscript.

*Figure 2: Labelled a) and b) but then not referenced in the caption as a) and b). Either have a* & *b in the caption or remove from figure.*

In other figures where there are 2 graphs, authors do not label them a) or b). Have a consistent format for the figures.

Uniformity in figure labels were maintained in the entire manuscript by removing a) and b) labels from Figure 2.

**References:**

Bruns, E. A., El Haddad, I., Keller, A., Klein, F., Kumar, N. K., Pieber, S. M., Corbin, J. C., Slowik, J. G., Brune, W. H., Baltensperger, U., and Prévôt, A. S. H.: Inter-comparison of laboratory smog chamber and flow reactor systems on organic aerosol yield and composition, *Atmos. Meas. Tech.*, *8*, 2315-2332, https://doi.org/10.5194/amt-8-2315-2015, 2015.

Statistic Report 2016, The Statistical Report 2016 of Flughafen Zürich AG is an addition to the Annual Report of Flughafen Zürich AG, 2016, available on https://www.zurich-airport.com/the-company/zurich-airport-ag/statistical-yearbook

Palm, B. B., Campuzano-Jost, P., Ortega, A. M., Day, D. A., Kaser, L., Jud, W., Karl, T., Hansel, A., Hunter, J. F., Cross, E. S., Kroll, J. H., Peng, Z., Brune, W. H., and Jimenez, J. L.: In situ secondary organic aerosol formation from ambient pine forest air using an oxidation flow reactor, *Atmos. Chem. Phys.*, *16*, 2943-2970, https://doi.org/10.5194/acp-16-2943-2016, 2016.

Stirnweis, L., Marcolli, C., Dommen, J., Barmet, P., Frege, C., Platt, S. M., Bruns, E. A., Krapf, M., Slowik, J. G., Wolf, R., Prévôt, A. S. H., Baltensperger, U., and El-Haddad, I.: Assessing the influence of NOx concentrations and relative humidity on secondary organic aerosol yields from  $\alpha$ -pinene photo-oxidation through smog chamber experiments and modelling calculations, *Atmos. Chem. Phys.*, *17*, 5035-5061, https://doi.org/10.5194/acp-17-5035-2017, 2017.

Yu, Z., Liscinsky, D.S., Winstead, E.L., True, B.S., Timko, M.T., Bhargava, A., Herndon, S.C., Miake-Lye, R.C. and Anderson, B.E., 2010. Characterization of lubrication oil emissions from aircraft engines. *Environ. Sci. Technol.*, 44(24), 9530-9534.

Yu, Z., Herndon, S.C., Ziemba, L.D., Timko, M.T., Liscinsky, D.S., Anderson, B.E. and Miake-Lye, R.C., 2012. Identification of lubrication oil in the particulate matter emissions from engine exhaust of in-service commercial aircraft. *Environ. Sci. Technol.*, *46*(17), 9630-9637.